# OPTIMAL BINARY QUANTIZATION FOR DEEP NEURAL NETWORKS

## ABSTRACT

Quantizing weights and activations of deep neural networks results in significant improvement in inference efficiency at the cost of lower accuracy. A source of the accuracy gap between full precision and quantized models is the quantization error. In this work, we focus on the binary quantization, in which values are mapped to -1 and 1. We introduce several novel quantization algorithms: optimal 1-bit, ternary, 2-bits, and greedy. Our quantization algorithms can be implemented efficiently on the hardware using bitwise operations. We present proofs to show that our proposed methods are optimal, and also provide empirical error analysis. We conduct experiments on the ImageNet dataset and show a reduced accuracy gap when using the proposed optimal quantization algorithms.

## 1 INTRODUCTION

A major challenge in the deployment of Deep Neural Networks (DNNs) is their high computational cost. Finding effective methods to improve run-time efficiency is still an area of research. We can group various approaches taken by researchers into the following three categories.

**Hardware optimization**: Specifically designed hardwares are deployed to efficiently perform computations in ML tasks. **Compiler optimization**: Compression and fusion techniques coupled with efficient hardware-aware implementations, such as dense and sparse matrix-vector multiplication, are used. **Model optimization**: Run-time performance can also be gained by modifying the model structure and the underlying arithmetic operations. While hardware and compiler optimization are typically lossless (i.e. incur no loss in model accuracy), model optimization trades-off computational cost (memory, runtime, or power) for model accuracy. For example, by scaling the width of the network (Zagoruyko & Komodakis, 2016). The goal of model optimization is to improve the trade-off between computational cost and model accuracy. This work falls into this category.

### 1.1 ARCHITECTURE OPTIMIZATION

One strategy to construct efficient DNNs is to define a template from which efficient computational blocks can be generated. Multiple instantiations of these blocks are then chained together to form a DNN. SqueezeNet (Iandola et al., 2016), MobileNets (Howard et al., 2017; Sandler et al., 2018), ShuffleNets (Zhang et al., 2018b; Ma et al., 2018), and ESPNets (Mehta et al., 2018; 2019) fall into this category. Complementary to these methods, NASNet (Zoph et al., 2018) and EfficientNet (Tan & Le, 2019) search for an optimal composition of blocks restricted to a computational budget (e.g., FLOPS) by changing the resolution, depth, width, or other parameters of each layer.

### 1.2 PRUNING AND COMPRESSION

Several methods have been proposed to improve runtime performance by detecting and removing computational redundancy. Methods in this category include low-rank acceleration (Jaderberg et al., 2014), the use of depth-wise convolution in Inception (Szegedy et al., 2015), sparsification of kernels in deep compression (Han et al., 2015), re-training redundant neurons in DSD (Han et al., 2016b), depth-wise separable convolution in Xception (Chollet, 2017), pruning redundant filters in PFA (Suau et al., 2018), finding an optimal sub-network in lottery ticket hypothesis (Frankle & Carbin, 2018), and separating channels based on the features resolution in octave convolution (Chen et al.,

2019). While some of these compression methods can be applied to a trained network, most add training-time constraints to create a computationally efficient model.

### 1.3 LOW-PRECISION ARITHMETIC AND QUANTIZATION

Another avenue to improve runtime performance (and the focus of this work) is to use low-precision arithmetic. The idea is to use fewer bits to represent weights and activations. Some instances of these strategies already exist in AI compilers, where it is common to cast weights of a trained model from 32 bits to 16 or 8 bits. However, in general, post-training quantization reduces the model accuracy. This can be addressed by incorporating lower-precision arithmetic into the training process (during-training quantization), allowing the resulting model to better adapt to the lower precision. For example, in Gupta et al. (2015); Jacob et al. (2018) the authors use 16 and 8 bits fixed-point representation to train DNNs.

Using fewer bits results in dramatic memory savings. This has motivated research into methods that use a single bit to represent a scalar weight: In Courbariaux et al. (2015) the authors train models with weights quantized to the values in $\{-1, 1\}$. While this results in a high level of compression, model accuracy can drop significantly. Li et al. (2016) and Zhu et al. (2016) reduce the accuracy gap between full precision and quantized models by considering ternary quantization (using the values in $\{-1, 0, 1\}$), at the cost of slightly less compression.

To further improve the computational efficiency, the intermediate activation tensors (feature maps) can also be quantized. When this is the case, an implementation can use high-performance operators that act on quantized inputs, for example a convolutional block depicted in Figure 1(left). This idea has been explored in (Courbariaux et al., 2016; Rastegari et al., 2016; Zhou et al., 2016; Hubara et al., 2017; Mishra et al., 2017; Lin et al., 2017; Cai et al., 2017; Ghasemzadeh et al., 2018; Zhang et al., 2018a; Choi et al., 2018), and many other works.

We call a mapping from a tensor with full precision entries to a tensor with the same shape but with values in $\{-1, 1\}$ a **binary quantization**. When both weights and activations of a DNN are quantized using binary quantization, called Binary Neural Network (BNN), fast and power-efficient kernels which use bitwise operations can be implemented. Observe that the inner-product between two vectors with entries in $\{-1, 1\}$ can be written as bitwise XNor operations followed by bit-counting (Courbariaux et al., 2016). However, the quantization of both weights and activations further reduces the model accuracy. In this work, we focus on improving the accuracy of the quantized model through improved quantization. The computational cost remains similar to the previous BNNs (Rastegari et al., 2016; Hubara et al., 2017).

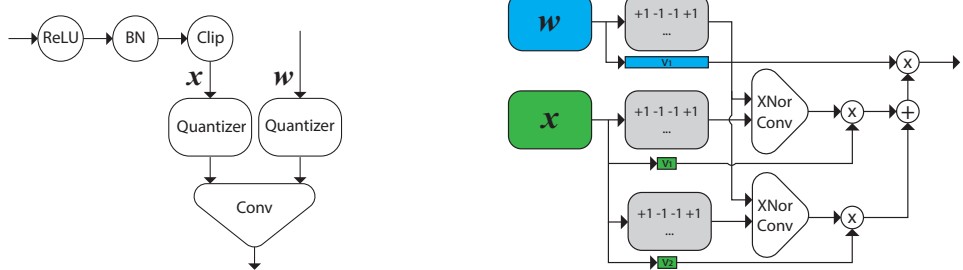

Figure 1: **Left**: The convolutional block used in this paper. **Right**: When both weights and activations are quantized using binary quantization, the convolution can be implemented efficiently using bitwise XNor and bit-counting operations. See Section 3.2 for more details.

### 1.4 MAIN CONTRIBUTIONS

In this work, we analyze the accuracy of binary quantization when applied to both weights and activations of a DNN, and propose methods to improve the quantization accuracy:

- We present an analysis of the quantization error and show that scaled binary quantization is a good approximation (Section 2).

- We derive the optimal 1-bit (Section 3.1.1), 2-bits (Section 3.2.2), and ternary (Section 3.2.3) scaled binary quantization algorithms.

- We propose a greedy $k$-bits quantization algorithm (Section 3.2.4).

- Experiments on the ImageNet dataset show that the optimal algorithms have reduced quantization error, and lead to improved classification accuracy (Section 5).

## 2 LOW-RANK BINARY QUANTIZATION

Binary quantization (that maps entries of a tensor to $\{-1, 1\}$) of weights and activation tensors of a neural network can significantly reduce the model accuracy. A remedy to retrieve this accuracy loss is to scale the binarized tensors with few full precision values. For example, Hubara et al. (2017) learn a scaling for each channel from the parameters of batch-normalization, and Rastegari et al. (2016) scale the quantized activation tensors using the channel-wise average of pixel values.

In this section, using low-rank matrix analysis, we analyze different scaling strategies. We conclude that multiplying the quantized tensor by a single scalar, which is computationally the most efficient option, has approximately the same accuracy as the more expensive alternatives.

We introduce the rank-1 binary quantization– an approximation to a matrix $\boldsymbol{X} \in \mathbb{R}^{m \times n}$:

$$\boldsymbol{X} \simeq \boldsymbol{X}_1 \odot \boldsymbol{S}, \tag{1}$$

where $\boldsymbol{X}_1 \in \mathbb{R}^{m \times n}$ is a rank-1 matrix, $\boldsymbol{S} \in \{-1, 1\}^{m \times n}$, and $\odot$ is element-wise multiplication (Hadamard product). Note that this approximation is also defined for tensors, after appropriate reshaping. For example, for an image classification task, we can reshape the output of a layer of a DNN with shape $h \times w \times n$, where $h$, $w$, and $n$ are height, width, and number of channels, respectively, into an $m \times n$ matrix with $m = hw$ rows and one column per channel.

We define the error of a rank-1 binary quantization as $\|\boldsymbol{X} - \boldsymbol{X}_1 \odot \boldsymbol{S}\|_F$, where $\| \|_F$ is the Frobenius norm. Entries of $\boldsymbol{S}$ are in $\{-1, 1\}$, therefore, the quantization error is equal to $\|\boldsymbol{X} \odot \boldsymbol{S} - \boldsymbol{X}_1\|_F$. Note that $\|\boldsymbol{X} \odot \boldsymbol{S}\|_F^2$ (the total energy), which is equal to sum of the squared singular values, is the same for any $\boldsymbol{S}$. Different choices of $\boldsymbol{S}$ change the distribution of the total energy among components of the Singular Value Decomposition (SVD) of $\boldsymbol{X} \odot \boldsymbol{S}$. The optimal rank-1 binary quantization is achieved when most of the energy of $\boldsymbol{X} \odot \boldsymbol{S}$ is in its first component.

In Rastegari et al. (2016), the authors proposed to quantize the activations by applying the sign function and scale them by their channel-wise average. We can formulate this scaling strategy as a special rank-1 binary quantization $\boldsymbol{X} \simeq \boldsymbol{a}\boldsymbol{1}^\top \odot \text{sign}(\boldsymbol{X})$, where

$$a_i = \frac{\sum_{j=1}^n |\boldsymbol{X}_{i,j}|}{n} \quad \text{for} \quad 1 \leq i \leq m \quad , \quad \text{sign}(x) = \begin{cases} -1 & \text{if } x < 0 \\ 1 & \text{if } x \geq 0 \end{cases}, \tag{2}$$

and $\boldsymbol{1}$ is an $n$-dimensional vector with all entries 1.

In Appendix A we show that the optimal rank-1 binary quantization is given by $\boldsymbol{S} = \text{sign}(\boldsymbol{X})$ and $\boldsymbol{X}_1 = \text{truncated}_1\text{-SVD}(|\boldsymbol{X}|)$, where $\text{sign}(\boldsymbol{X})$ is the element-wise sign of $\boldsymbol{X}$, and $\text{truncated}_1\text{-SVD}(|\boldsymbol{X}|) = \sigma_1 \boldsymbol{u}_1 \boldsymbol{v}_1^\top$ is the first component of the SVD of $\boldsymbol{X} \odot \text{sign}(\boldsymbol{X}) = |\boldsymbol{X}|$. Moreover, we empirically analyze the accuracy of the optimal rank-1 binary quantization for a random matrix $\boldsymbol{X}$, where its entries are i.i.d. $\sim \mathcal{N}(0, 1)$. This is a relevant example since after the application of Batch Normalization (BN) (Ioffe & Szegedy, 2015) activation tensors are expected to have a similar distribution. The first singular value of $|\boldsymbol{X}|$ captures most of the energy $\sigma_1^2(|\boldsymbol{X}|)/\|\boldsymbol{X}\|_F^2 \simeq 0.64$, and the first left and right singular vectors are almost constant vectors. Therefore, a scalar multiple of $\text{sign}(\boldsymbol{X})$ approximates $\boldsymbol{X}$ well: $\boldsymbol{X} \simeq \sigma_1 \boldsymbol{u}_1 \boldsymbol{v}_1^\top \odot \text{sign}(\boldsymbol{X}) \simeq v \, \boldsymbol{1}\boldsymbol{1}^\top \odot \text{sign}(\boldsymbol{X}) = v\text{sign}(\boldsymbol{X})$, where $v \in \mathbb{R}_{\geq 0}$. We use this computationally efficient approximation called **scaled binary quantization**.

## 3 SCALED BINARY QUANTIZATION

In Section 2 we showed that scaled binary quantization is a good approximation to activation and weight tensors of a DNN. Next we show how we can further improve the accuracy of scaled binary quantization using more bits. To simplify the presentation (1) we flatten matrix $\boldsymbol{X} \in \mathbb{R}^{m \times n}$ in to

a vector $\boldsymbol{x} \in \mathbb{R}^N$ with $N = mn$, and (2) we assume the entries of $\boldsymbol{x}$ are different realizations of a random variable x with an underlying probability distribution $p(x)$. In practice, we compute all statistics using their unbiased estimators from vector $\boldsymbol{x}$ (e.g., $\sum_i x_i/N$ is an unbiased estimator of $\mathbb{E}_{x \sim p}[x]$). Furthermore, for $f : \mathbb{R} \rightarrow \mathbb{R}$, we denote entrywise application of $f$ to $\boldsymbol{x}$ by $f(\boldsymbol{x})$. The quantized approximation of $\boldsymbol{x}$ is denoted by $\boldsymbol{x}^q$, and the error (loss) of quantization is $\|\boldsymbol{x} - \boldsymbol{x}^q\|_2$. All optimal solutions are with respect to this error and hold for an arbitrary distribution $p(x)$.

## 3.1 1-BIT QUANTIZATION

A 1-bit scaled binary quantization of $\boldsymbol{x}$ is:

$$\boldsymbol{x} \simeq \boldsymbol{x}^q = vs(\boldsymbol{x}), \tag{3}$$

which is determined by a scalar $v \in \mathbb{R}_{\geq 0}$ and a function $s : \mathbb{R} \rightarrow \{-1, 1\}$. Finding the optimal 1-bit scaled binary quantization can be formulated as the following optimization problem:

$$\begin{aligned} \underset{v,s}{\text{minimize}} \quad & \int_{-\infty}^{+\infty} p(x)(v\,s(x) - x)^2 dx \\ \text{s.t.} \quad & s : \mathbb{R} \rightarrow \{-1, 1\}, v \in \mathbb{R}_{\geq 0} \end{aligned} \tag{4}$$

### 3.1.1 OPTIMAL 1-BIT ALGORITHM

The solution of problem (4) is given by $v = \mathbb{E}_{x \sim p}[|x|]$ and $s(x) = \text{sign}(x)$ (for the proofs see Appendix B). Therefore, for a vector $\boldsymbol{x}$ the optimal scaled binary quantization is given by

$$\boldsymbol{x} \simeq \boldsymbol{x}^q = \frac{\sum_i |x_i|}{N} \text{sign}(\boldsymbol{x}), \tag{5}$$

where $\frac{\sum_i |x_i|}{N}$ is an unbiased estimator of $\mathbb{E}_{x \sim p}[|x|]$.

## 3.2 $k$-BITS QUANTIZATION

We can further improve the accuracy of scaled binary quantization by adding more terms to the approximation (3). A $k$-bits scaled binary quantization of $\boldsymbol{x}$ is

$$\boldsymbol{x} \simeq \boldsymbol{x}^q = \sum_{i=1}^{k} v_i s_i(\boldsymbol{x}), \tag{6}$$

which is determined by a set of $k$ pairs of scalars $v_i$'s and functions $s_i : \mathbb{R} \rightarrow \{-1, 1\}$. Observe that any permutation of $(v_i, s_i)$'s results in the same quantization. To remove ambiguity, we assume $v_1 \geq \ldots \geq v_k \geq 0$.

When both weights, $\boldsymbol{w}$, and activations, $\boldsymbol{x}$, are quantized using scaled binary quantization (6), their inner-product can be written as:

$$\langle \boldsymbol{x}^q, \boldsymbol{w}^q \rangle = \sum_{i=1}^{k^a} \sum_{j=1}^{k^w} v_i^a v_j^w \langle \boldsymbol{s}_i^a, \boldsymbol{s}_j^w \rangle, \tag{7}$$

where $\boldsymbol{x}^q = \sum_{i=1}^{k^a} v_i^a \boldsymbol{s}_i^a$ and $\boldsymbol{w}^q = \sum_{j=1}^{k^w} v_j^w \boldsymbol{s}_j^w$ are quantized activations and weights with $k^a$ and $k^w$ bits, respectively, $\boldsymbol{s}_i^a = s_i^a(\boldsymbol{x})$, and $\boldsymbol{s}_j^w = s_j^w(\boldsymbol{w})$. This inner-product can be computed efficiently using bitwise XNors followed by bit-counting (see Figure 1(right) with $k^a = 2$ and $k^w = 1$).

Finding the optimal $k$-bits scaled binary quantization can be formulated as:

$$\begin{aligned} \underset{s_i,v_i}{\text{minimize}} \quad & \int_{-\infty}^{+\infty} p(x) \left( \left( \sum_{i=1}^{k} v_i s_i(x) \right) - x \right)^2 dx \\ \text{s.t.} \quad & \forall\, 1 \leq i \leq k \quad s_i : \mathbb{R} \rightarrow \{-1, 1\}, \quad v_1 \geq v_2 \geq \ldots \geq v_k \geq 0 \end{aligned} \tag{8}$$

This is an optimization problem with a non-convex domain for all $k \geq 1$. We solve the optimization for $k = 1$ in Section 3.1 and $k = 2$ in Section 3.2.2 for arbitrary distribution $p(x)$. We also provide an approximate solution to (8) in Section 3.2.4 using a greedy algorithm.

**Discussion:** A general $k$-bits quantizer maps full precision values to an arbitrary set of $2^k$ numbers, not necessarily in the form of (6). The optimal quantization in this case can be computed using the Lloyd's algorithm (Lloyd, 1982). While a general $k$-bits quantization has more representation power compared to $k$-bits scaled binary quantization, it does not allow an efficient implementation based on bitwise operations. Fixed-point representation (as opposed to floating point) is also in the form of (6) with an additional constant term. However, fixed-point quantization uniformly quantizes the space, therefore, it can be significantly inaccurate for small values of $k$.

### 3.2.1 FOLDABLE QUANTIZATION

In this section, we introduce a special family of $k$-bits scaled binary quantizations that allow fast computation of the quantized values. We name this family of quantizations **foldable**. A $k$-bits scaled binary quantization given by $(v_i, s_i)$'s is foldable if the following conditions are satisfied:

$$s_i(x) = \text{sign}(x - \sum_{j=1}^{i-1} v_j s_j(x)) \quad \text{for } 1 \le i \le k \tag{9}$$

When the foldable condition is satisfied, given $v_i$'s, we can compute the $s_i(\boldsymbol{x})$'s in (6) efficiently by applying the sign function.

### 3.2.2 OPTIMAL 2-BITS ALGORITHM

In this section, we present the optimal 2-bits binary quantization algorithm, the solution of (8) for $k = 2$. In Appendix C we show that the optimal 2-bits binary quantization is foldable and the scalars $v_1$ and $v_2$ should satisfy the following optimality conditions:

$$v_1 = \frac{1}{2} \left( \mathbb{E}_{\text{x} \sim p}[|\text{x}| \mid |\text{x}| > v_1] + \mathbb{E}_{\text{x} \sim p}[|\text{x}| \mid |\text{x}| \le v_1] \right) \tag{10}$$

$$v_2 = \frac{1}{2} \left( \mathbb{E}_{\text{x} \sim p}[|\text{x}| \mid |\text{x}| > v_1] - \mathbb{E}_{\text{x} \sim p}[|\text{x}| \mid |\text{x}| \le v_1] \right) \tag{11}$$

In Figure 2 we visualize the conditional expectations that show up in (10) for a random variable x with standard normal distribution. The optimal $v_1$ lies on the intersection of the identity line and average of the conditional expectations in (10).

For a given vector $\boldsymbol{x} \in \mathbb{R}^N$ we can solve for $v_1$ in (10) efficiently. We substitute the conditional expectations in (10) by conditional average operators as their unbiased estimators. (10) implies that for the optimal $v_1$, the average of the entries in $|\boldsymbol{x}|$ smaller than $v_1$ (an estimator of $\mathbb{E}_{\text{x} \sim p}[|\text{x}| \mid |\text{x}| \le v_1]$ ) and the average of the entries greater than $v_1$ (an estimator of $\mathbb{E}_{\text{x} \sim p}[|\text{x}| \mid |\text{x}| > v_1]$) should be equidistant form $v_1$. Note that (10) may have more than one solution, which are local minima of the objective function in (8). We find all the values that satisfy this condition in $\mathcal{O}(N \log N)$ time. We first sort entries of $\boldsymbol{x}$ based on their absolute value and compute their cumulative sum. Then with one pass we can check whether (10) is satisfied for each element of $\boldsymbol{x}$. We evaluate the objective function in (8) for each local minima, and retain the best. After $v_1$ is calculated $v_2$ is simply computed from (11). As explained in Section 4, this process is only done during the training. In our experiments, finding the optimal 2-bits quantization increased the training time by 35% compared to the 2-bits greedy algorithm (see Section 3.2.4). Sine the optimal 2-bits binary quantization is foldable, after recovering $v_1$ and $v_2$, we have $s_1(\boldsymbol{x}) = \text{sign}(\boldsymbol{x})$ and $s_2(\boldsymbol{x}) = \text{sign}(\boldsymbol{x} - v_1\text{sign}(\boldsymbol{x}))$.

### 3.2.3 OPTIMAL TERNARY ALGORITHM

The optimization domain of (8) for $k = 2$ over the scalars is illustrated in Figure 2(right). The boundaries of the domain, $v_2 = 0$ and $v_1 = v_2 = v$, correspond to 1-bit binary and ternary (Li et al., 2016) quantizations, respectively. The scaled ternary quantization maps each full precision value $x$ to $\{-2v, 0, 2v\}$. Ternary quantization needs 2-bits for representation. However, when a hardware with sparse calculation support is available, for example as in EIE (Han et al., 2016a), using ternary quantization can be more efficient compared to general 2-bits quantization. In Appendix D we show that the optimal scaled ternary quantization is foldable and the scalar $v$ should satisfy:

$$v = \frac{1}{2} \mathbb{E}_{\text{x} \sim p}[|\text{x}| \mid |\text{x}| > v] \tag{12}$$

The process of solving for $v$ in (12) is similar to that of solving for $v_1$ in (10) as described above.

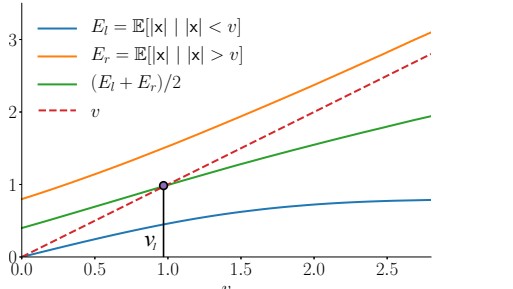 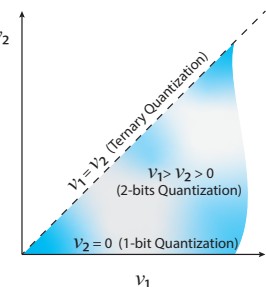

Figure 2: **Left**: The conditional expectations in (10) for a random variable x with standard normal distribution. The optimal value for 2-bits quantization is shown with a solid dot. **Right**: Optimization domain of (8) for k=2. The boundaries correspond to 1-bit and ternary quantizations.

### 3.2.4  $k$-BITS GREEDY ALGORITHM

In this section, we propose a greedy algorithm to compute $k$-bits scaled binary quantization, which we call Greedy Foldable (GF). It is given in Algorithm 1.

---

**Algorithm 1:** $k$-bits Greedy Foldable (GF) binary quantization: compute $\boldsymbol{x}^q$ given $\boldsymbol{x}$

---
$\boldsymbol{r} \leftarrow \boldsymbol{x}$
**for** $i \leftarrow 1$ **to** $k$ **do**
 $\quad v_i \leftarrow \text{mean}(\text{abs}(\boldsymbol{r}))$
 $\quad \boldsymbol{s}_i \leftarrow \text{sign}(\boldsymbol{r})$ // element-wise sign.  For gradient of sign use STE.
 $\quad \boldsymbol{r} \leftarrow \boldsymbol{r} - v_i \boldsymbol{s}_i$ // compute new residual.
**end**
**return** $\boldsymbol{x} - \boldsymbol{r}$

---

In GF algorithm we compute a sequence of residuals. At each step, we greedily find the best $\boldsymbol{s}_i$ and $v_i$ for the current residual using the optimal 1-bit binary quantization (5). Note that for $k = 1$ the GF is the same as the optimal 1-bit binary quantization.

Few of the other papers that have tackled the $k$-bits binary quantization to train quantized DNNs are as follows. In ReBNet (Ghasemzadeh et al., 2018), the authors proposed an algorithm similar to Algorithm 1, but considered $v_i$'s as trainable parameters to be learned by back-propagation. Lin et al. (2017) and Zhang et al. (2018a) find $k$-bits binary quantization via alternating optimization for $\boldsymbol{s}_i$'s and $v_i$'s. Note that, all these methods produce sub-optimal solutions.

## 4  TRAINING BINARY NETWORKS

The loss functions in our quantized neural networks are non-differentiable due to the sign function in the quantizers. To address this challenge we use the training algorithm proposed in Courbariaux et al. (2015). To compute the gradient of the sign function we use the Straight Through Estimator (STE) (Bengio et al., 2013): $d/dx \, \text{sign}(x) = \mathbf{1}_{|x| \leq 1}$. During the training we keep the full precision weights and use Stochastic Gradient Descent (SGD) to gradually update them in back-propagation. In the forward-pass, only the quantized weights are used.

During the training we compute quantizers (for both weights and activations) using the online statistics, i.e., the scalars in a $k$-bits scaled binary quantization (6) are computed based on the observed values. During the training we also store the running average of these scalars. During inference we use the stored quantized scalars to improve the efficiency. This procedure is similar to the update of the batch normalization parameters in a standard DNN training (Ioffe & Szegedy, 2015).

## 5  EXPERIMENTS

We conduct experiments on the ImageNet dataset (Deng et al., 2009) using the ResNet-18 architecture (He et al., 2016). The details of the architecture and training are provided in Appendix E.

We conduct three sets of experiments: (1) evaluate quantization error of activations of a pre-trained DNN, (2) evaluate the quantization error based on the classification accuracy of a post-training quantized network, and (3) evaluate the classification accuracy of during-training quantized networks. We report the quantization errors of the proposed binary quantization algorithms (optimal 1-bit, 2-bits, ternary, and the greedy foldable quantizations) and compare with the state-of-the-art algorithms BWN-Net (Rastegari et al., 2016), XNor-Net (Rastegari et al., 2016), TWN-Net (Li et al., 2016), DoReFa-Net (Zhou et al., 2016), ABC-Net (Lin et al., 2017), and LQ-Net (Zhang et al., 2018a).

### 5.1 QUANTIZATION ERROR OF ACTIVATIONS

To quantify the errors of the introduced binary quantization algorithms we adopt the analysis performed by Anderson & Berg (2017). They show that the angle between $x$ and $x^q$ can be used as a measure of the accuracy of a quantization scheme. They prove that when $x^q = \text{sign}(x)$ and elements of $x$ are i.i.d. $\sim \mathcal{N}(0, 1)$, $\angle(x, x^q)$ converges to $\sim 37$ degrees for large $N$.

Here we use the real data distribution. We trained a full precision network. We compute the activation tensors at each layer for a set of 128 images. In Figure 3 we show the angle between the full precision and quantized activations for different layers. When the optimal quantization is used, a significant reduction in the angle is observed compared to the greedy algorithm. The optimal 2-bits quantization is even better than the greedy 4-bits quantization for later layers of the network, for which activation tensors have more skewed distribution, make it harder for quantization in form of (6). Furthermore, the accuracy of the optimal quantization has less variance with respect to different input images and different layers of the network.

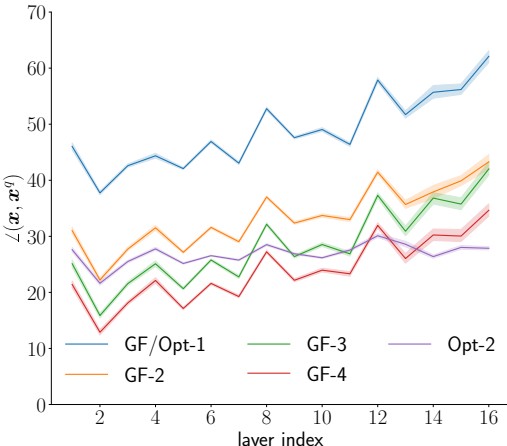

Figure 3: The angle between the full precision and the quantized activations for different layers of a trained full precision ResNet-18 architecture on ImageNet. The 95% confidence interval over different input images is shown.

| Method | $k^a$ | $k^w$ | Top-1 | Top-5 |
|---|---|---|---|---|
| Post-GF | 32 | 1 | 0.1 | 0.5 |
| Post-GF | 32 | 2 | 0.3 | 1.1 |
| Post-GF | 32 | 3 | 1.4 | 4.6 |
| Post-GF | 32 | 4 | 5.3 | 14.1 |
| Post-Opt | 32 | 2 | 5.3 | 13.9 |
| Opt | 1 | 1 | 54.3 | 77.3 |
| GF | 2 | 1 | 59.7 | 81.7 |
| GF | 3 | 1 | 61.1 | 82.7 |
| GF | 4 | 1 | 61.3 | 82.8 |
| Opt | T | 1 | 58.3 | 80.6 |
| Opt | 2 | 1 | 60.4 | 82.2 |
| FP | 32 | 32 | 69.6 | 89.2 |

Table 1: Validation accuracy of a quantized ResNet-18 trained on ImageNet. $k^a$ and $k^w$ are number of bits to quantize activations and weights, respectively. T, Opt, GF, and FP refer to ternary, optimal, Greedy Foldable, and full precision, respectively.

### 5.2 POST-TRAINING QUANTIZATION

In this section we apply post-training quantization to the weights of a pre-trained full precision network. We then use the quantized network for inference and report the classification accuracy. This procedure can result in an acceptable accuracy for a moderate number of bits (e.g., 16 or 8). However, the error significantly grows with a lower number of bits, which is the case in this experiment. Therefore, we only care about the relative differences between different quantization strategies. This experiment demonstrates the effect of quantization errors on the accuracy of the quantized DNNs.

The results are shown in the top half of Table 1. When the optimal 2-bits quantization is used, significant accuracy improvement (more than one order of magnitude) is observed compared to the greedy 2-bits quantization, which illustrate the effectiveness of the optimal quantization.

## 5.3 DURING-TRAINING QUANTIZATION

To achieve higher accuracy we apply quantization during the training, so that the model can adapt to the quantized weights and activations. In the bottom half of Table 1, we report the accuracies of the during-training quantized DNNs, all trained with the same setup. We use 1-bit binary quantization for weights, and use different quantization algorithms for activations. When quantization is applied during-training, significantly higher accuracies are achieved. Similar to the previous experiments the optimal quantization algorithm achieves a better accuracy compared to the greedy.

In Table 2 we report results from the related works in which ResNet-18 architecture with quantized weights and/or activations is trained on the ImageNet dataset for the classification task. We report the mean and standard deviation of the model accuracy over 5 runs when our algorithms are used. Note that for 1-bit quantization the Greedy Foldable (GF) algorithm is the same with the optimal 1-bit binary quantization. In Opt* we used $2\times$ larger batch-size compared to Opt but with the same number of optimization steps. As shown in the Table 2 the proposed quantization algorithms match or improve the accuracies of the state-of-the-art BNNs.

| Method (ResNet-18 on ImageNet) | $k^a$ | $k^w$ | Val. top-1 | Val. top-5 |
|---|---|---|---|---|
| XNor-Net (Rastegari et al., 2016) | 1 | 1 | 51.2 | 73.2 |
| Opt | 1 | 1 | $54.3 \pm 0.2$ | $77.4 \pm 0.2$ |
| Opt | T | 1 | $58.3 \pm 0.1$ | $80.6 \pm 0.0$ |
| DoReFa-Net (Zhou et al., 2016)[a] | 2 | 1 | 53.4 | - |
| LQ-Net (Zhang et al., 2018a) | 2 | 1 | 62.6 | 84.3 |
| HWGQ-Net (Cai et al., 2017) | 2 | 1 | 59.6 | 82.2 |
| GF | 2 | 1 | $59.7 \pm 0.2$ | $81.7 \pm 0.1$ |
| Opt | 2 | 1 | $60.4 \pm 0.2$ | $82.2 \pm 0.1$ |
| Opt* | 2 | 1 | $62.4 \pm 0.1$ | $83.6 \pm 0.1$ |
| GF | 3 | 1 | $61.1 \pm 0.1$ | $82.7 \pm 0.1$ |
| ABC-Net (Lin et al., 2017) | 3 | 3 | 61.0 | 83.2 |
| DoReFa-Net (Zhou et al., 2016) | 4 | 1 | 59.2 | 81.5 |
| GF | 4 | 1 | $61.3 \pm 0.2$ | $82.8 \pm 0.1$ |
| BWN-Net (Rastegari et al., 2016) | 32 | 1 | 60.8 | 83.0 |
| Opt | 32 | 1 | $64.2 \pm 0.4$ | $85.2 \pm 0.1$ |
| TWN-Net (Li et al., 2016) | 32 | T | 61.8 | 84.2 |
| Opt | 32 | T | $64.4 \pm 0.1$ | $85.4 \pm 0.1$ |
| FP | 32 | 32 | 69.6 | 89.2 |

Table 2: Comparison with state-of-the-art quantization. Opt and GF are the proposed optimal and greedy foldable quantization algorithms, respectively. T and FP refer to ternary and full precision network, respectively.

---

[a]This result is taken from (Zhang et al., 2018a).

## 6 CONCLUSION

In this work, we analyze the accuracy of binary quantization to train DNNs with quantized weights and activations. We discuss methods to improve the accuracy of quantization, namely scaling and using more bits.

We introduce the rank-1 binary quantization, as a general scaling scheme. Based on a singular value analysis we motivate using the scaled binary quantization, a computationally efficient scaling strategy. We define a general $k$-bits scaled binary quantization. We provide provably optimal 1-bit, 2-bits, and ternary quantizations. In addition, we propose a greedy $k$-bits quantization algorithm. We show results for post and during-training quantization, and demonstrate significant improvement in accuracy when optimal quantization is used. We compare the proposed quantization algorithms with state-of-the-art BNNs on the ImageNet dataset and show improved classification accuracies.

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

## A    OPTIMAL RANK-1 BINARY QUANTIZATION

In this section, we find the optimal rank-1 binary quantization of an $m$ by $n$ matrix $\boldsymbol{X}$ that is discussed in Section 2:

$$
\begin{aligned}
\underset{\boldsymbol{X}_1, \boldsymbol{S}}{\text{minimize}} \quad & \|\boldsymbol{X} - \boldsymbol{X}_1 \odot \boldsymbol{S}\|_F \\
\text{s.t.} \quad & \boldsymbol{S} \in \{-1, 1\}^{m \times n} \\
& \boldsymbol{X}_1 \in \mathbb{R}^{m \times n} \\
& \text{rank}(\boldsymbol{X}_1) = 1
\end{aligned}
\tag{13}
$$

First, observe that the element-wise multiplication by $-1$ and $+1$ does not change the Frobenius norm. Therefore:

$$
\min_{\boldsymbol{S}, \boldsymbol{X}_1} = \|\boldsymbol{X} - \boldsymbol{X}_1 \odot \boldsymbol{S}\|_F = \min_{\boldsymbol{S}, \boldsymbol{X}_1} = \|(\boldsymbol{X} - \boldsymbol{X}_1 \odot \boldsymbol{S}) \odot \boldsymbol{S}\|_F = \min_{\boldsymbol{S}, \boldsymbol{X}_1} \|\boldsymbol{X} \odot \boldsymbol{S} - \boldsymbol{X}_1\|_F \tag{14}
$$

Furthermore, note that

$$
\min_{\boldsymbol{S}, \boldsymbol{X}_1} \|\boldsymbol{X} \odot \boldsymbol{S} - \boldsymbol{X}_1\|_F^2 = \sigma_2^2(\boldsymbol{X} \odot \boldsymbol{S}) + \ldots + \sigma_r^2(\boldsymbol{X} \odot \boldsymbol{S}) \tag{15}
$$

Here $\sigma_i(\boldsymbol{X} \odot \boldsymbol{S})$ is the $i$'th singular value of $\boldsymbol{X} \odot \boldsymbol{S}$ and $r$ is its rank. In addition for any $\boldsymbol{S}$:

$$
\sum_{i=1}^{r} \sigma_i^2(\boldsymbol{X} \odot \boldsymbol{S}) = \|\boldsymbol{X} \odot \boldsymbol{S}\|_F^2 = \|\boldsymbol{X}\|_F^2 \tag{16}
$$

Hence, to minimize the sum in (15) we need to find an $\boldsymbol{S}$ for which $\sigma_1^2(\boldsymbol{X} \odot \boldsymbol{S})$ is maximized:

$$
\min_{\boldsymbol{S}, \boldsymbol{X}_1} \|\boldsymbol{X} \odot \boldsymbol{S} - \boldsymbol{X}_1\|_F^2 = \|\boldsymbol{X}\|_F^2 - \max_{\boldsymbol{S}} \sigma_1^2(\boldsymbol{X} \odot \boldsymbol{S}) \tag{17}
$$

$\sigma_1(\boldsymbol{X} \odot \boldsymbol{S}) = \|\boldsymbol{X} \odot \boldsymbol{S}\|_2$ is the 2-norm of $\boldsymbol{X} \odot \boldsymbol{S}$. Therefore:

$$
\max_{\boldsymbol{S}} \sigma_1^2(\boldsymbol{X} \odot \boldsymbol{S}) = \max_{\boldsymbol{S}} \max_{\|\boldsymbol{r}\|_2 = 1} \|(\boldsymbol{X} \odot \boldsymbol{S})\boldsymbol{r}\|_2^2 \tag{18}
$$

For any $\boldsymbol{S}$ and $\boldsymbol{r} \in \mathbb{R}^n$ we have $\|(\boldsymbol{X} \odot \boldsymbol{S})\boldsymbol{r}\|_2^2 \leq \||\boldsymbol{X}||\boldsymbol{r}|\|_2^2$ since for $1 \leq i \leq m$ we have $|\sum_j S_{i,j} X_{i,j} r_j| \leq \sum_j |X_{i,j}||r_j|$. Here $|\boldsymbol{X}| = \boldsymbol{X} \odot \text{sign}(\boldsymbol{X})$ is the element-wise absolute value of $\boldsymbol{X}$. Note that for $\boldsymbol{S} = \text{sign}(\boldsymbol{X})$ and $\boldsymbol{r}$ with positive values the inequality becomes an equality. Therefore:

$$
\max_{\boldsymbol{S}} \max_{\|\boldsymbol{r}\|_2 = 1} \|(\boldsymbol{X} \odot \boldsymbol{S})\boldsymbol{r}\|_2^2 = \max_{\|\boldsymbol{r}\|_2 = 1} \||\boldsymbol{X}||\boldsymbol{r}|\|_2^2 \tag{19}
$$

Observe that the element-wise absolute value does not change the vector norm, i.e. $\||\boldsymbol{r}|\|_2 = \|\boldsymbol{r}\|_2$, and hence $|\boldsymbol{r}|$ is a unit vector when $\boldsymbol{r}$ is. Also for any $\boldsymbol{r}$ we have $\||\boldsymbol{X}|\boldsymbol{r}\|_2^2 \leq \||\boldsymbol{X}||\boldsymbol{r}|\|_2^2$ since for $1 \leq i \leq m$ we have $|\sum_j |X_{i,j}| r_j| \leq \sum_j |X_{i,j}||r_j|$. So we have

$$
\max_{\|\boldsymbol{r}\|_2 = 1} \||\boldsymbol{X}||\boldsymbol{r}|\|_2^2 = \max_{\|\boldsymbol{r}\|_2 = 1} \||\boldsymbol{X}|\boldsymbol{r}\|_2^2 = \sigma_1^2(|\boldsymbol{X}|) \tag{20}
$$

Therefore, we showed that $\boldsymbol{S} = \text{sign}(\boldsymbol{X})$ and $X_1$ equal to the best rank-1 approximation of $|\boldsymbol{X}|$ (i.e. the first term in its SVD) is a solution of (13).

For an $\boldsymbol{X}$ with i.i.d. $\sim \mathcal{N}(0, 1)$ entries we show the singular values and first left and right singular vectors of $|\boldsymbol{X}|$ in Figure 4. Observe that the first singular value of $|\boldsymbol{X}|$ captures most of the energy $\sigma_1^2(|\boldsymbol{X}|)/\|\boldsymbol{X}\|_F^2 \simeq 0.64$. The fraction of energy captured by the first component of SVD converges to the squared mean of the standard folded normal distribution $(2/\pi)$ for large square matrices. Also note that the first left and right singular vectors are almost constant, i.e., they can be written as $\alpha\boldsymbol{1}$ for some $\alpha \in \mathbb{R}$.

## B    OPTIMAL SCALED BINARY QUANTIZATION

In this section, we solve the following optimization problem corresponding to the optimal 1-bit scaled binary quantization as discussed in Section 3.1:

$$
\begin{aligned}
\underset{v, s}{\text{minimize}} \quad & \int_{-\infty}^{+\infty} p(x)(v\, s(x) - x)^2 dx \\
\text{s.t.} \quad & s : \mathbb{R} \to \{-1, 1\} \\
& v \in \mathbb{R}_{\geq 0}
\end{aligned}
\tag{21}
$$

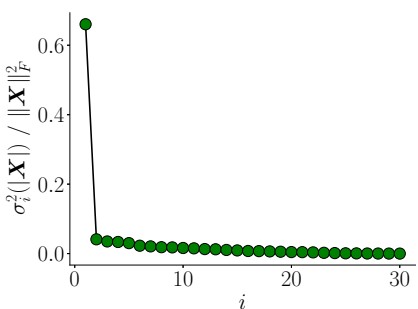 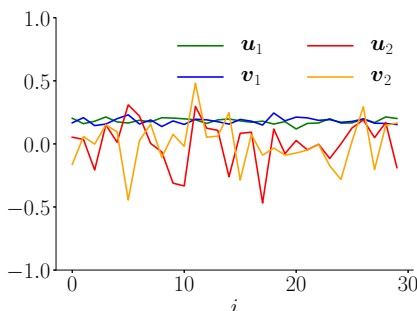

Figure 4: **Left**: Distribution of energy for $|X|$, where $X \in \mathbb{R}^{30 \times 30}$ is a standard normal random matrix. **Right**: Entries of the first left and right singular vectors of $|X|$ (shown in green and blue) are almost constant.

Here $p$ is a probability distribution function. First, observe that:

$$\forall x \in \mathbb{R} : (v - x)^2 < (-v - x)^2 \quad \text{iff} \quad x > 0 \tag{22}$$

Therefore, the optimal choice for function $s$ is $s(x) = \text{sign}(x)$. So we can rewrite (21) as follows:

$$\underset{v}{\text{minimize}} \quad \int_{-\infty}^{+\infty} p(x)(v - |x|)^2 dx \tag{23}$$
$$\text{s.t.} \quad v \in \mathbb{R}_{\geq 0}$$

Setting the gradient of the objective function in (23) with respect to $v$ to zero, we get:

$$v = \frac{\int_{-\infty}^{+\infty} p(x)|x|dx}{\int_{-\infty}^{+\infty} p(x)dx} = \mathbb{E}_{\mathrm{x} \sim p}[|\mathrm{x}|] \tag{24}$$

Hence, we showed that the optimal 1-bit scaled binary quantization maps $x$ to $\mathbb{E}_{\mathrm{x} \sim p}[|\mathrm{x}|] \, \text{sign}(x)$.

## C  OPTIMAL 2-BITS BINARY QUANTIZATION

In this section, we solve the following optimization problem corresponding to the optimal 2-bits binary quantization as discussed in Section 3.2.2:

$$\underset{v_1,v_2,s_1,s_2}{\text{minimize}} \quad \int_{-\infty}^{+\infty} p(x) \left(v_1 s_1(x) + v_2 s_2(x) - x\right)^2 dx \tag{25}$$
$$\text{s.t.} \quad s_1, s_2 : \mathbb{R} \to \{-1, 1\}$$
$$v_1 \geq v_2 \geq 0$$

First, we show that the optimal 2-bits binary quantization is foldable, i.e., $\forall x \in \mathbb{R} \; s_1(x) = \text{sign}(x)$ and $s_2(x) = \text{sign}(x - v_1 s_1(x))$. Observe that

$$f(x) = (v_1 s_1(x) + v_2 s_2(x) - x)^2 = v_1^2 \left(1 + \frac{v_2}{v_1} s_1(x) s_2(x) - \frac{s_1(x)x}{v_1}\right)^2 \geq$$
$$v_1^2 \left(1 + \frac{v_2}{v_1} s_1(x) s_2(x) - \frac{|s_1(x)x|}{v_1}\right)^2 = g(x) \tag{26}$$

The inequality in (26) holds because $v_1 \geq v_2$, and therefore, $1 + \frac{v_2}{v_1} s_1(x) s_2(x) \geq 0$. The objective function in (25) is a weighted average of $f(x)$ with non-negative weights. For $x \in \mathbb{R}$ the inequality is strict if $s_1(x) \neq \text{sign}(x)$. In that case, flipping the value of both $s_1(x)$ and $s_2(x)$ reduces $f(x)$ to a strictly smaller value $g(x)$. Hence, the optimal solution of (25) should satisfy $s_1(x) = \text{sign}(x)$ for all $x \in \mathbb{R}$.

For any $v_1$ and $s_1$ if we consider $y = x - v_1 s_1(x)$, the problem reduces to the 1-bit binary quantization for $y$. Based on the result showed in Appendix B for the optimal solution we have $s_2(x) = \text{sign}(y) = \text{sign}(x - v_1 s_1(x))$. This completes the proof to show that the optimal 2-bits binary quantization is foldable.

Next, we find the optimal values for $v_1$ and $v_2$. If we substitute $s_1(x) = \text{sign}(x)$ and $s_2(x) = \text{sign}(x - v_1 s_1(x))$ in (25) we can decompose $\mathbb{R}$ into four segments and write:

$$
\begin{aligned}
e(v_1, v_2) = \int_{-\infty}^{+\infty} p(x) \left(v_1 s_1(x) + v_2 s_2(x) - x\right)^2 dx = \\
\int_{-\infty}^{-v_1} p(x)(x + v_1 + v_2)^2 dx + \int_{-v_1}^{0} p(x)(x + v_1 - v_2)^2 dx + \\
\int_{0}^{v_1} p(x)(x - v_1 + v_2)^2 dx + \int_{v_1}^{+\infty} p(x)(x - v_1 - v_2)^2 dx = \\
\int_{0}^{v_1} q(x)(x - v_1 + v_2)^2 dx + \int_{v_1}^{+\infty} q(x)(x - v_1 - v_2)^2 dx
\end{aligned}
\tag{27}
$$

Here $e(v_1, v_2)$ is the error as a function of $v_1$ and $v_2$, and $q(x) = p(-x) + p(x)$ is the folded distribution function. Assuming the optimal point occurs in the interior of the domain, it should satisfy the zero gradient condition: $\partial e / \partial v_1 = \partial e / \partial v_2 = 0$. Taking derivative from (27) with respect to $v_1$ and $v_2$ and set it to zero we get:

$$
\begin{aligned}
v_1 = \int_{0}^{v_1} x q(x) dx + \int_{v_1}^{+\infty} x q(x) dx \quad + v_2 \left( \int_{0}^{v_1} q(x) dx - \int_{v_1}^{+\infty} q(x) dx \right) \\
v_2 = - \int_{0}^{v_1} x q(x) dx + \int_{v_1}^{+\infty} x q(x) dx \quad + v_1 \left( \int_{0}^{v_1} q(x) dx - \int_{v_1}^{+\infty} q(x) dx \right)
\end{aligned}
\tag{28}
$$

We can simplify (28) and rewrite $v_1$ and $v_2$ in terms of the following conditional expectations:

$$
\begin{aligned}
v_1 = \frac{1}{2} \left( \mathbb{E}_{x \sim p}[|x| \mid |x| > v_1] + \mathbb{E}_{x \sim p}[|x| \mid |x| \le v_1] \right) \\
v_2 = \frac{1}{2} \left( \mathbb{E}_{x \sim p}[|x| \mid |x| > v_1] - \mathbb{E}_{x \sim p}[|x| \mid |x| \le v_1] \right)
\end{aligned}
\tag{29}
$$

Hence, the optimal values of $v_1$ and $v_2$ can be obtained by solving for $v_1$ and $v_2$ in (29). The optimization domain has two boundaries. One is when $v_2 = 0$. This reduces the problem to 1-bit binary quantization. The optimal solution in that case is discussed in Appendix B. The other boundary is when $v_1 = v_2$. This results in the ternary quantization. The optimal solution in this case is discussed in Appendix D.

## D  OPTIMAL TERNARY QUANTIZATION

In this section, we find the optimal symmetric ternary quantization, that is to map a full precision value $x$ to a discrete set $\{-2v, 0, 2v\}$ as discussed in Section 3.2.3. Finding the optimal mapping can be formulated as the following optimization problem:

$$
\begin{aligned}
\underset{v, s_1, s_2}{\text{minimize}} \quad & \int_{-\infty}^{+\infty} p(x) \left(v s_1(x) + v s_2(x) - x\right)^2 dx \\
\text{s.t.} \quad & s_1, s_2 : \mathbb{R} \to \{-1, 1\} \\
& v \ge 0
\end{aligned}
\tag{30}
$$

First, the above form is a special case of (25), hence, using the same argument as in Appendix C we can show that there is a foldable optimal solution: $s_1(x) = \text{sign}(x)$ and $s_2(x) = \text{sign}(x - v s_1(x))$. Then the total error as a function of $v$ can be written as:

$$
e(v) = \int_{0}^{v} q(x) x^2 dx + \int_{v}^{+\infty} q(x)(x - 2v)^2 dx
\tag{31}
$$

where $q(x)$ is the folded probability distribution function. Taking derivative from (31) and setting it to zero, we get:

$$
v = \frac{\int_{v}^{+\infty} x q(x) dx}{2 \int_{v}^{+\infty} q(x) dx} = \frac{1}{2} \mathbb{E}_{x \sim p}[|x| \mid |x| > v]
\tag{32}
$$

## E  DETAILS OF TRAINING RESNET ON IMAGENET

In this section, we explain the details of how the DNN results reported in this paper are produced. All results correspond to the ResNet-18 architecture trained on the ImageNet dataset for the classification task. We use the standard training and validation splits of the ImageNet dataset. We followed a similar architecture as XNor-Net (Rastegari et al., 2016). The convolutional block that we use is depicted in Figure 1(left). We use ReLU non-linearity before the batch normalization as suggested by (Rastegari et al., 2016). Also, we find it important to use bounded dynamic range, and therefore clip the values to $[-d, d]$. We use $d = 2$, 3, 5, and 8 for $k = 1$, 2, 3, and 4 bits quantizations, respectively. Similar to the other BNNs for the first and last layers we use full precision. Also, as suggested by Choi et al. (2018) we use full precision short-cuts in ResNet architecture, which adds a small computational/memory overhead. We quantize weights per filter and activations per layer. As Cai et al. (2017) we use first-order polynomial learning-rate annealing schedule (from $10^{-1}$ to $10^{-4}$) and train for 120 epochs. We do not use weight decay. For the data augmentation we use the standard methods used to train full precision ResNet architecture. For training we apply random resize and crop to 224×224, followed by random horizontal flipping, color jittering, and lightening. For test we resize the images to 256×256 followed by a center cropping to 224×224.

