# OpenReview forum: "OPTIMAL BINARY QUANTIZATION FOR DEEP NEURAL NETWORKS"
_ICLR.cc/2020/Conference — Reject_

### Official Review · AnonReviewer3 · 2019-10-09
**Official Blind Review #3**

**Rating:** 3

**Review:**

this paper looks at different quantization schemes (replace values of vector in \R to, basically, plus or minus v, for well chosen v).

Different schemes are considered, namely the low rank binary quantization (a matrix is approximated by the componentwise product of a low rank matrix and a +/-1 matrix) and k-bit binary  quantization.
Finding the best quantization ends up in solving a program, which is convex and quite straightforward fo k=1 and 2, and unfortunately (apparently) non-convex for k>2. So the authors suggest a greedy approach for k>2.

The motivation of this work is DNN, arguing that quantized vectors should improve computations cost, hence some experiments are provide on DNN, yet they only illustrate the fact that quantization does not deteriorate too much the learning & test error.

The main criticisms is therefore the lack of concrete evidences that those schemes are actually helpful and, on the other hand, the relative simplicity (so the theoretical part of the paper are not sufficient by itself)

**Experience Assessment:**

I do not know much about this area.

**Review Assessment: Checking Correctness Of Derivations And Theory:**

I assessed the sensibility of the derivations and theory.

**Review Assessment: Checking Correctness Of Experiments:**

I did not assess the experiments.

**Review Assessment: Thoroughness In Paper Reading:**

I read the paper at least twice and used my best judgement in assessing the paper.

---

> ### Author Response · Authors · 2019-11-13
> **Response to Review 3**
>
> We would like to thank the reviewer for their time and helpful feedback. We believe that the concern of the reviewer is due to a misunderstanding (convexity of the optimization problem) which we clarify below. We would appreciate if the reviewer would reevaluate the contributions based on the provided clarifications.
>
> Comment 1: (“Finding the best quantization ends up in solving a program, which is convex and quite straightforward for k=1 and 2, and unfortunately (apparently) non-convex for k>2 ...”)
> The optimization problem discussed in this paper is not convex for any $k$. In (8), the optimization is over both $s_i$’s (functions with range in {-1,1}) and $v_i$’s (non-negative scalars). This optimization domain is a non-convex set. For this non-convex problem, we derive the optimal solution when $k$ is equal to 1, 2 and for ternary quantization. The solution to this problem requires finding a solution that satisfies equations (10) and (11). An efficient algorithm is explained in Section 3.2.2 to solve equations (10) and (11). We suppose the convexity discussion after equation (8) might have led to the misunderstanding that the problem is convex for $k=1$ and 2, therefore, in the revised draft we modified this section  to be more clear.
>
> Comment 2: (“The motivation of this work is DNN, arguing that quantized vectors should improve computations cost, hence some experiments are provide on DNN ...”)
> This work is a follow-up on a sequence of papers on Binary Neural Networks (BNNs). In this work, we specifically focussed on improving the quantization accuracy to improve the model performance, while maintaining a similar computational saving as previous works (for example in [1] the authors show 23x speed-up in matrix-matrix multiplication when binary quantization is used, in [2] the authors report 58x speed up when using BNNs). Therefore, we did not discuss latency and power saving when XNOR based computations are used. We clarified this in the revised draft.
>
> Comment 3: (“The main criticisms is therefore the lack of concrete evidences that those schemes are actually helpful and, on the other hand, the relative simplicity ...”)
> We respectfully disagree this criticism.
> Experiments: Our goal in this paper is to improve the quantization accuracy, while keeping the same computational budget as original BNN/XNor papers. In Section 5, we present a comprehensive set of experiments: (1) quantization error of activations, (2) post-training quantization, and (3) during-training quantization. All experiments show that the proposed optimal quantization algorithms significantly improve the accuracy.
> Theory:  Introduced rank-1 binary quantization analysis is a unified framework to analyze different scaling strategies that were proposed in the literature (e.g. scaling used in XNOR-net [2]). Using this analysis, we show that the scaled binary quantization is a good approximation to a full precision tensor. Moreover, as discussed above, the optimization problem (8) is non-convex. For this non-convex problem, we derive the provably optimal solution when $k$ is equal to 1, 2 and for ternary quantization for an arbitrary data distribution.
>
> [1] Hubara, Itay, et al. "Quantized neural networks: Training neural networks with low precision weights and activations." The Journal of Machine Learning Research 18.1 (2017): 6869-6898.
> [2] Mohammad Rastegari, Vicente Ordonez, Joseph Redmon, and Ali Farhadi. Xnor-net: Imagenet classification using binary convolutional neural networks. In European Conference on Computer Vision, pp. 525–542. Springer, 2016.

---

### Official Review · AnonReviewer2 · 2019-10-21
**Official Blind Review #2**

**Rating:** 6

**Review:**

This paper proposes a new family of quantized neural networks, where the weights are quantized based on fixed precisions. The authors proposed the optimal 1-bit/2-bit/ternary quantization schemes based on minimizing the L_2 loss. The authors proposed a greedy algorithm to approximate the optimal k-bit quantization.  The authors showed through extensive experiments on real DNNs that the proposed optimal quantization (Opt in the tables) can give better generalization as compared to several state-of-the-art alternative methods.

The reviewer appreciates that this quantized DNN-paper has a theory. This theory is based on minimizing the L_2 loss between the original data and quantized data. It unifies several quantization schemes into one framework. This is in contract with technical papers in the area which propose a single quantization method. The proposed quantization can further be implemented efficiently using XNOR operations and bit-counting.

The writing is of good quality. The length is a bit larger than the recommended length of 8 pages.

The reviewer has the following concerns,

- This paper motivates from rank-k quantization but implemented as a scaled quantization. At the end of page 3, the authors showed that the scaled quantization can somehow approximate the rank-k quantization. However, the approximation is loose without any guarantee. Overall, I don't understand how the low-rank quantization in section 2 is related to the proposed method and fits in the overall picture. Ideally, in section 2, the authors can have some theoretical statements to state the optimality of the rank-k quantization. Then, the authors can say that, because of practical difficulties to implement the SVD, the use the scaled binary quantization instead. Or, the authors can simply remove section 2, and add a paragraph to introduce rank-k quantization, which is in contract to the scaled quantization they used in the paper.

Somewhere in the text, the authors have to explain the optimality is with respect to the L_2 loss. There can be alternative quantization based on different losses.

Figure 4, could you explain why the angle trends up as the layer index increases?

After equation(4): mention the choice of p(x)

In conclusion, analyz -> analyze, introduc->introduce

**Experience Assessment:**

I have read many papers in this area.

**Review Assessment: Checking Correctness Of Derivations And Theory:**

I assessed the sensibility of the derivations and theory.

**Review Assessment: Checking Correctness Of Experiments:**

I assessed the sensibility of the experiments.

**Review Assessment: Thoroughness In Paper Reading:**

I read the paper thoroughly.

---

> ### Author Response · Authors · 2019-11-13
> **Response to Review 2**
>
> We would like to thank the reviewer for their time and constructive feedback. We are glad that the reviewer appreciated the significance of the presented theory. The following are our responses to reviewer’s comments.
>
> Comment 1: (“The writing is of good quality. The length is a bit larger than the recommended length of 8 page”)
> As suggested by the reviewer we compressed the manuscript to 8 pages.
>
> Comment 2: (Relation of rank-k quantization in Section 2 to the proposed method)
> As reviewer mentioned computing the optimal low-rank binary quantization through SVD is computationally expensive. Our goal to introduce low rank binary quantization is providing a unified framework to analyze different scaling strategies used in the literature. For example, we show that the scaling used in XNOR-net is a special rank-1 binary quantization. Moreover, using this framework we show that “scaled binary quantization” is a good approximation to a full precision tensor with i.i.d. Normal entries (which is approximately the case in layer activations or weights of a full precision neural network). In our derivation we first approximate a full precision tensor by the optimal rank-1 binary quantization. We then show that the optimal rank-1 binary quantization captures most of the energy (~0.64) of the full precision matrix, and can be approximated as a scalar times a binary matrix.
>
> Thanks for the suggestion to simplify Section 2. In the revised manuscript, we removed the rank-k quantization and only discuss the rank-1 binary quantization to make the above points clearer. This also helped us to compress the paper to 8 pages as suggested by the reviewer.
>
> Comment 3: (“Somewhere in the text, the authors have to explain the optimality is with respect to the L_2 loss”)
>  In the revised draft, we clarified the choice of L2 loss and the optimal solution depends on the loss (in the first paragraph of Section 3).
>
> Comment 4: (“Figure 4, could you explain why the angle trends up as the layer index increases”)
> In Figure 4 (Figure 3 in the revised manuscript), we investigate the accuracy of different quantization schemes discussed in the manuscript for activation maps of a trained (non-quantized) network. The network is a trained image classifier, therefore, as the layer index increases the feature maps become more linearly separable. This is done through the composition of several layers of non-linearities (ReLU in this case). Application of ReLU in each layer results in concentration of many inactive feature maps at zero. Hence, after batch-normalization we observe more skewed distribution in later layers, in contrast to early layers that are more similar to normal distribution. The family of binary quantization schemes discussed in this paper (that enables efficient implementation using bit-wise XNOR operations) are symmetric, hence, are less accurate for more skewed distributions. Figure 4 (Figure 3 in the revised manuscript) shows that when the introduced optimal 2-bits quantization is used, the sensitivity of error (angle) to layer index is significantly less. We clarified this in the revised draft.
>
> Comment 5: (“After equation(4): mention the choice of p(x)”)
> The optimal quantizations introduced in Section 3 (1-bit, 2-bits, and ternary) hold for any distribution. We do not make any specific choice for distribution p(x). We clarified this in the revised draft in the first paragraph of Section 3.
>
> Comments 6: (Typos)
> Thanks for pointing out to the typos. We corrected them in the revised manuscript.

---

> > ### Comment · AnonReviewer2 · 2019-11-13
> > **From rank-1 quantization to scaled quantization**
> >
> > Thank you for the revision, which has improved the presentation.
> >
> > I still don't understand why rank-1 quantization can be approximated by scaled quantization. In your approximation, why u_1, v_1, the singular vectors can be approximated as a vector of all 1's?

---

> > > ### Author Response · Authors · 2019-11-14
> > > **On the connection between rank-1 binary quantization and scaled binary quantization**
> > >
> > > Thank you for your prompt feedback.
> > > We make this conclusion by the following steps:
> > >
> > > 1) For any matrix $X$ we prove in Appendix A that the best (in Frobenius norm sense) rank-1 binary quantization is the truncated SVD of its element-wise absolute value matrix, shown by truncated-SVD-1($|X|$), times (Hadamard) its element-wise sign.
> > >
> > > 2) We then consider a case where entries of $X \in \mathbb{R}^{m \times n}$ are i.i.d. $\sim \mathcal{N}(0,1)$, a case that is close to the actual distribution of weights and activations (after batch normalization) in neural networks.
> > >
> > > 3) With the assumption (2), we empirically show (in Figure 4 in the revised manuscript) that the truncated-SVD-1($|X|$) $\approx \mu {\bf 1} {\bf 1}^T$ where $\mu = \sqrt{2/\pi}$. We did not present a rigorous proof since it involves asymptotic analysis of largest singular values of a random matrix with non-zero mean which we believed to be not very relevant to the contributions of the paper and would unnecessarily complicate it. The sketch of this proof is as follows:  In [1] and [2] the authors show that the largest singular value of a random matrix with i.i.d. entries from a distribution with mean $\mu$ and bounded 4th moment (which is the case for standard folded normal distribution) asymptotes to $\sqrt{mn}\mu$ as $m$ and $n$ are increased (with $m/n \to$constant). Note that $\mathbb{E}[\frac{{\bf 1}^T}{\sqrt{m}} |X|~\! |X|^T \frac{{\bf 1}}{\sqrt{m}} ]  = \mathbb{E}[\frac{{\bf 1}^T}{\sqrt{n}} |X|^T ~\! |X| \frac{{\bf 1}}{\sqrt{n}} ] = mn\mu^2$ (with convergence given by the central limit theorem), and therefore, for large matrices the first left and right singular vectors are expected to be almost constant. If the reviewer believes that this proof adds to the paper, we will be happy to include it.
> > >
> > > From (3), we conclude that: (a) The optimal rank-1 binary quantization captures $2/\pi \simeq 0.64$ of the total energy of $X$, making it a good approximation, (b) optimal rank-1 binary quantization can be written as a scalar times a binary matrix.
> > >
> > >
> > > [1] Silverstein, Jack W. "The spectral radii and norms of large dimensional non-central random matrices." Stochastic Models 10.3 (1994): 525-532.
> > > [2] Bryc, Wlodek, and Jack W. Silverstein. "Singular values of large non-central random matrices." arXiv preprint arXiv:1802.02960 (2018).

---

### Official Review · AnonReviewer1 · 2019-10-23
**Official Blind Review #1**

**Rating:** 6

**Review:**

This work proves seval binary quantization guarantee for the 1 or 2 bit cases, and shows some empirical error analysis.

I am not an expert on neural network compression so I am not quite sure how the proposed method compares with the state-of-the-art algorithms. On the other hand, I checked several proofs provided by the authors for the 1-bit and 2-bit quantization cases. The proofs look good to me.

Some minor comments:
1. For the definition (9) can the authors make it clear that it is for all v_j s.t. v_1>= v_2>=.... instead of \exits v_j?
2. Can authors provide some explanation why in (8) we want to have v1>=v2>=vk in the constraint? I understand we need that in the proof, but is there any reason this is also the case in empirical evaluation? To me we may also have cases such that v1 < v2, is there any guarantee for those cases?
3. Feels like the draft can be compressed into 8 pages, even if the work looks nice.


**Experience Assessment:**

I have read many papers in this area.

**Review Assessment: Checking Correctness Of Derivations And Theory:**

I assessed the sensibility of the derivations and theory.

**Review Assessment: Checking Correctness Of Experiments:**

I assessed the sensibility of the experiments.

**Review Assessment: Thoroughness In Paper Reading:**

I read the paper at least twice and used my best judgement in assessing the paper.

---

> ### Author Response · Authors · 2019-11-13
> **Response to Review 1**
>
> We would like to thank the reviewer for their time and helpful feedback. We particularly appreciate checking the proofs in the Appendix. The following are our responses to reviewer’s comments.
>
> Comments 1&2: (Clarification on $v_1 \ge v_2 \ge \ldots \ge v_k \ge 0$ constraint)
> Any permutation of ($v_i$, $s_i$)’s results in the same k-bits quantization. To remove ambiguity we assume that $v_1 \ge v_2 \ge \ldots \ge v_k \ge 0$. Note that this constraint only removes the ambiguity but we still consider all the possible k-bits quantizations. Thanks for pointing to a potential confusion. We clarified this in the revised draft after equation (6) and before equation (9).
>
> Comment 3: (“Feels like the draft can be compressed into 8 pages, even if the work looks nice“)
> As suggested we compressed the draft into 8 pages.
>
> The network architecture we use in the paper is a standard binary network (slightly improved implementation of [1]) and close to the state-of-the-art (see Table 2). Our goal in this work is on improving quantization for binary neural networks. In terms of quantization, the proposed methods are provably optimal. We empirically demonstrate that optimal quantization leads to improved accuracy on this strong baseline. We note that, there are other directions to further improve the accuracy of BNNs (architecture optimization, training methods, hyper parameters, etc.), all complementary to the proposed optimal quantization.
>
> [1] Mohammad Rastegari, Vicente Ordonez, Joseph Redmon, and Ali Farhadi. Xnor-net: Imagenet classification using binary convolutional neural networks. In European Conference on Computer Vision, pp. 525–542. Springer, 2016.

---

### Official Review · AnonReviewer4 · 2019-11-24
**Official Blind Review #4**

**Rating:** 3

**Review:**

This paper proposes to quantize the weights of neural networks that can minimize the L_2 loss between the quantized values and the full-precision ones. The authors propose solutions for optimal 1-bit/ternary/2-bit quantization, as well as a greedy algorithm to approximate the optimal k-bit quantization.  Experiments are performed on image classification data set ImageNet using ResNet-18.

First of all, the authors should explicitly state in the paper that the optimality is in terms of what?  Indeed the authors obtain the solution of (1) quantization with a scaling parameter, and (2) in terms of minimizing quantization error (L_2 loss between the quantized value and the full-precision one), which is quite restricted to be a universally optimal one.

Since the number of weights and activations are limited in the network, it is not appropriate to formulate quantization error the weights and activations in the format of continuous distribution in (4) and (8).

It is also not clear to me why this paper begins with rank-1 quantization but ends up with scaled quantization. What kind of assumption are used in this approximation, and can the optimality still be guaranteed?

One of my main concerns is the novelty of this paper. Many of the solutions in the paper have already been discovered in literature. For instance, the optimal 1-bit solution in (5) was already obtained in Binary-Weight-Network [1] in 2016. The optimal ternary solution (i.e., the ternarization threshold should be 1/2 of the scaling parameter) in (12)  was also already obtained in Corollary 3.1 in  [2] in 2018, as a special case when curvature information is dropped.

Yet another concern is about the experiments. Since the proposed optimal binarization has the same solution as BWN, where does the performance gain in Table 2 come from? Moreover, some of the recent quantization methods are not compared. For instance, in PACT [3], 2-bit weight&acitvation quantization already achieves 64.4% top-1 accuracy of Resnet18 on ImageNet, while the proposed method achieves the same accuracy with full-precision activation and 2-bit weight (Table 2). In addition, according to Table 2, the proposed method also can not beat LQ-Net.


[1] Rastegari, Mohammad, et al. "Xnor-net: Imagenet classification using binary convolutional neural networks." ECCV, 2016.
[2] Hou, Lu, and James T. Kwok. "Loss-aware weight quantization of deep networks." ICLR 2018.
[3] Choi, Jungwook, et al. "Pact: Parameterized clipping activation for quantized neural networks." arXiv preprint arXiv:1805.06085 (2018).

**Experience Assessment:**

I have published in this field for several years.

**Review Assessment: Checking Correctness Of Derivations And Theory:**

I assessed the sensibility of the derivations and theory.

**Review Assessment: Checking Correctness Of Experiments:**

I assessed the sensibility of the experiments.

**Review Assessment: Thoroughness In Paper Reading:**

I read the paper at least twice and used my best judgement in assessing the paper.

---

> ### Author Response · Authors · 2020-01-15
> **Response to Review 4**
>
> We would like to thank the reviewer for their time and valuable comments. However, we would like to note that, the comments would be more helpful if were posted before the end of discussion period to find a constructive solution.
>
> Comment 1: ("the authors should explicitly state in the paper that the optimality is in terms of what...")
> The optimality refers to the least squares solution of quantization of isolated tensors (both weights and activations). It is mentioned at the beginning of page 4 (Section 3) in the paper: “All optimal solutions are with respect to this error and hold for an arbitrary distribution p(x). ”
>
> Simultaneous quantization of all layers , or loss aware quantization techniques may result in better model accuracy compared to independent quantization. However, In this paper we study the effect of quantization error in isolation and propose provably optimal (in L2 sense) 2-bits Xnor quantization.
>
> Comment 2: ("... it is not appropriate to formulate quantization error the weights and activations in the format of continuous distribution...")
> We respectfully disagree with the reviewer. The number of weights and activations for networks of interests (e.g., resnet18) are statistically large, make the continuous approximation accurate enough. We have verified this by sub-sampling tensors (i.e., use a fraction of values in a tensor instead of all its elements) and observe negligible change in the resulted statistics. Moreover, as mentioned in the paper continuous notation is used only for ease of derivation and explanation of the methodologies.
>
> Comment 3: ("It is also not clear to me why this paper begins with rank-1 quantization...")
> Please see our answer to Reviewer 2 regarding this question.
>
> Comment 4: ("One of my main concerns is the novelty of this paper...")
> The main novelty of the paper is on rank-1 analysis and optimal 2-bits quantization solution. The optimal 1-bit quantization is provided for completeness, and optimal ternary is mentioned as a specific case of 2-bits quantization. We thank the reviewer to point to [2], which unfortunately we missed in our previous works review. As the reviewer mentioned, in [2] the optimal ternary quantization is obtained when Hessian is replaced by scalar times identity matrix; similarly, we obtain optimal ternary quantization when add a constraint on scaling factors in our general 2-bits solution: v1 and v2 to be equal.
>
> Comment 5: ("Yet another concern is about the experiments...")
> The performance boost in our implementation of BWN comes from using full precision short-cuts. It is mentioned in Appendix E in training details.
>
> PACT uses uniform quantization with 2-bits for weights and activations. In this paper we focus on networks with binary weights, 1-bit per value (therefore maximum size compression), and improve model accuracy by reducing the quantization error of  activations. Hence, results are not directly comparable. Please note that the adaptive clipping mechanism introduced in PACT is orthogonal to our optimal quantization techniques and could be used together.
>
> We propose provably optimal 2-bits quantization and support optimality of the method empirically as well. The final accuracy depends on other factors as well (e.g., learning rate schedule and batchsize as shown in Table 2), that we did not tune for. LQ-Net provides a suboptimal solution to the optimization problem that we solve in this paper.  Few other differences: LQ-Net uses quantization with {0,1} encoding for activations and {-1,1} for weights (while we use {-1,1} for both), and uses a different order of operations in a layer for network than Xnor-Net (that we adopted in this paper). Our final accuracy results for ImageNet-ResNet18 are close (62.4% vs 62.6%).

---

### Decision · Program_Chairs · 2019-12-19

**Decision:**

Reject

**Comment:**

This paper proposes to quantize the weights of neural networks that can minimize the L_2 loss between the quantized values and the full-precision ones. The paper has limited novelty, as many of the solutions presented in the paper have already been discovered in the literature. During the discussion, the reviewers agree that it is an incremental contribution. Parts of the paper can also be clarified, particularly on the optimality of the solution, assumptions used in the approximation, and some of the experimental results. Experimental results can also be made more convincing by adding comparision with the more recent quantization methods.